# RNA Helicase DDX3 Interacts with the Capsid Protein of Hepatitis E Virus and Plays a Vital Role in the Viral Replication

**DOI:** 10.3390/pathogens14020177

**Published:** 2025-02-10

**Authors:** Shaoli Lin, Bhargava Teja Sallapalli, Peixi Chang, Jia He, Etienne Coyaud, Brian G. Pierce, Yan-Jin Zhang

**Affiliations:** 1Molecular Virology Laboratory, Department of Veterinary Medicine, University of Maryland, College Park, MD 20742, USA; linshaoli19903@gmail.com (S.L.); sbteja@umd.edu (B.T.S.); changpxi@umd.edu (P.C.); hejia86@hotmail.com (J.H.); 2U1192-Protéomique Réponse Inflammatoire Spectrométrie de Masse (PRISM), CHU Lille, National Institute of Health and Medical Research (INSERM), Universite de Lille, F-59000 Lille, France; 3Institute for Bioscience and Biotechnology Research, University of Maryland, Rockville, MD 20850, USA; pierce@umd.edu; 4Department of Cell Biology and Molecular Genetics, University of Maryland, College Park, MD 20742, USA

**Keywords:** hepatitis E virus (HEV), interferon (IFN), capsid protein, DDX3, RNA helicase

## Abstract

DDX3 is an ATP-dependent RNA helicase that is involved in multiple cellular activities, including RNA metabolism and innate immunity. DDX3 is known to assist the replication of some viruses while restricting others through its direct interaction with viral proteins. However, the role of DDX3 in the replication of the hepatitis E virus (HEV) is unknown. In this study, DDX3 was shown to interact with the HEV capsid protein and provide an important role in HEV replication. The DDX3 C-terminal domain was demonstrated to interact with the capsid protein. The depletion of DDX3 led to a significant reduction in HEV replication. Also, the ATPase motif of DDX3 was shown to be required in HEV replication as an ATPase-null mutant DDX3 failed to rescue the viral replication in the DDX3-depleted cells. These results demonstrate a pro-viral role of DDX3 in HEV replication, providing further insights on the virus–cell interactions.

## 1. Introduction

Helicases are divided into six superfamilies on the basis of sequence homology in conserved motifs [1]. The DEAD (Asp-Glu-Ala-Asp)-box RNA helicases are members of the superfamily 2 and are ubiquitously expressed in all eukaryotes and most prokaryotes [2,3]. DEAD-box helicase 3, X-linked (DDX3X), is involved in almost all aspects of RNA metabolism, from RNA transcription, splicing, transport, and translation to decay. DDX3X is also involved in innate immunity, viral infection, and tumorigenesis [3,4,5,6]. DDX3X is located on the X chromosome and is present in all human tissues. The deletion of DDX3X in male mice is embryonic-lethal [7]. DDX3X has a paralog, DDX3Y, which is located on the Y chromosome and is closely related to male fertility [8]. It is observed that DDX3Y is frequently lost in patients with infertility [9].

DDX3X (hereinafter referred to as DDX3) consists of two RecA-like domains, domain 1 (aa166–407) and 2 (aa414–576). Mutation of the lysine 230 residue within the N-terminal ATPase motif of DDX3 results in the loss of ATPase activity. Mutation of the serine 382 residue within the helicase motif abolishes the RNA unwinding activity of DDX3 [10]. DDX3 couples with the mitochondrial antiviral-signaling protein (MAVS) to promote the transcription of interferons (IFNs). Depletion of DDX3 reduces IFN production [11]. Previous studies have demonstrated that DDX3 is required for viral replication in some viruses [3,12]. During the replication of hepatitis C virus (HCV), DDX3 is recruited to the lipid droplets, where HCV virions assemble. Depletion of DDX3 in hepatoma cells significantly inhibits HCV replication [13]. DDX3 promotes arenavirus replication by enhancing viral RNA synthesis. DDX3 with mutations of either the ATPase motif (K230E) or the helicase motif (S382A) fails to promote arenavirus RNA synthesis [14]. In contrast to the findings above, DDX3 can exert an inhibitory role in some other viruses. DDX3 interacts with influenza virus NS1 and NP via its C-terminal domain and is associated with stress granules to inhibit viral infection [15]. Silencing DDX3 attenuates the formation of stress granules and also enhances viral replication. DDX3 is also shown to inhibit HBV replication by interacting with polymerase and incorporating itself into the nucleocapsid. The ATPase activity of DDX3 is essential in viral inhibition [16].

Hepatitis E virus (HEV) is a single-stranded positive-sense RNA virus that is one of the most common causes of acute hepatitis. HEV infection can cause fulminant hepatic failure, leading to a high case-fatality rate among infected pregnant women in south Asian countries [17,18,19]. Acute HEV infection also causes substantial morbidity in pregnant women in high-income countries [20]. HEV strains are heterogenic and there are four genera in the subfamily *Orthohepevirinae*, the *Hepeviridae* family [21]. There are eight genotypes in the species *Paslahepevirus balayani*, the genus *Paslahepevirus*. At least four of the eight genotypes can infect humans: genotypes 1 and 2 are restricted to humans, whereas genotypes 3 and 4 are zoonotic with an expanded host range, including monkey, pig, sheep, cow, wild boar, deer, rabbit, and mongoose [22,23]. Genotypes 5 and 6 are isolated from wild boars [22] and genotypes 7 and 8 are isolated from camels [24,25]. Strains of genotypes 5 and 7 are suspected of having the potential for zoonotic infection [25,26]. There are three open reading frames (ORFs) in the HEV genome: ORF1 encodes the non-structural polyprotein for viral RNA replication, ORF2 encodes the capsid protein, and ORF3 encodes a multifunctional protein [21,23].

The capsid protein is demonstrated to inhibit IFN induction via the blocking of the phosphorylation of interferon regulatory factor 3 (IRF3) [27]. The objective of this study was to determine if other factors are involved in the inhibition and the role of DDX3 in HEV replication. Here, DDX3 was found to be associated with the HEV capsid protein. Also, the depletion of DDX3 inhibits HEV replication, suggesting that DDX3 plays an important role in viral infection. A mutagenesis analysis shows that the ATPase motif of DDX3 is required for HEV replication. These results reveal the critical role of DDX3 in HEV proliferation and provide insights into the virus-cell interactions.

## 2. Materials and Methods

### 2.1. Cells and Viruses

Huh7.5.1 [28], GP2-293 (Clontech Laboratories, Inc., Mountain View, CA, USA), HEK293T (ATCC^®^ CRL-3216^™^, Manassas, VA, USA), and HepG2/C3A (ATCC^®^ CRL-3581) cells were cultured in Dulbecco’s modified Eagle’s medium (DMEM, Corning, Corning, NY, USA) supplemented with 10% fetal bovine serum (FBS, Minneapolis, MN, USA). In vitro transcription was performed from p6/luc, a HEV replicon that contains cDNA of the genomic RNA of Kernow-C1, a genotype 3 strain, with an insert encoding Gaussia luciferase, replacing the 5′ portion of ORF2 [29]. The luciferase is secreted out of the cells after synthesis. The Kernow-C1 strain p6 was used to infect the HepG2/C3A cells at a multiplicity of infection (MOI) of 1. The infected HepG2/C3A cells were passaged five times to produce stably infected cells.

### 2.2. Plasmids

The ORF2 from the HEV Kernow-C1 strain (GenBank Accession Number: JQ679013) was cloned to the Myc-BioID2-MCS vector (a gift from Kyle Roux, Addgene plasmid # 74223; http://n2t.net/addgene:74223 (accessed on 20 December 2024); RRID:Addgene_74223) [30] for the expression of a fusion protein, and to the pCAGEN vector with a HA-tag. The DDX3 ORF clone (NM_001356) was purchased from OriGene Technologies (Rockville, MD, USA). DDX3 was amplified using PCR for subcloning to the pCAGEN vector with a FLAG-tag at the N-terminus, as described previously [27]. The DDX3 mutants (K230E and S382L) were generated using overlapping PCR and ligated to the pCAGEN vector with a HA-tag at the N-terminus [27]. DDX3 D1 and D2 truncations were ligated to the pCAGEN vector with a HA-tag at the C-terminus. The oligos for shRNA against DDX3 (Table 1) were cloned to the pSIREN-RetroQ-ZsGreen vector as instructed (Clontech, Mountain View, CA, USA). The primers used in this study are listed in Table 1. All the plasmids constructed in-house were subjected to verification using Sanger DNA sequencing.

### 2.3. Transfection

The transfection of cells with plasmid DNA was performed using the jetOPTIMUS^®^ DNA transfection reagent (Polyplus, New York, NY, USA) according to the instructions of the manufacturer. Low molecular weight poly(I:C) (Invivogen, San Diego, CA, USA), a synthetic analog of double-stranded RNA (dsRNA), was used to induce interferon production. GP2-293 cells were transfected with poly(I:C) at a concentration of 1 µg/mL overnight before being harvested for further analysis.

### 2.4. Immunofluorescence Assay (IFA)

The IFA was performed as reported previously [31] with primary antibodies against the Myc-tag (ABclonal, Woburn, MA, USA) and the FLAG-tag (Sigma-Aldrich, St. Louis, MO, USA) and fluorescein-conjugated secondary antibodies: goat anti-rabbit IgG(H&L) Dylight 594 and goat anti-mouse IgG(H&L) Dylight 488 (Proteintech Group, Inc., Rosemont, IL, USA). ProLong Antifade Mountants containing DAPI (4=,6=-diamidino-2phenylindole) (Thermo Fisher Scientific, Waltham, MA, USA) were used for the coverslip mounting. The imaging was conducted using a Zeiss LSM800 confocal microscope and Zeiss Zen imaging software (version 2.6) (Carl Zeiss AG, Jena, Germany). The co-localization of two fluorophores was measured using the software to obtain the Pearson’s correlation coefficient (PCC): a value of 1 indicates perfect correlation, 0 for no correlation, and -1 for ideal anti-correlation.

### 2.5. DDX3 Depletion

For the depletion of DDX3 in GP2-293 cells, the cells were co-transfected with pSIREN-RetroQ-ZsGreen-shDDX3 and pVSV-G plasmids at a ratio of 1:1. At 72 h later, the cell lysates were collected for DDX3 detection using a western blot. For the DDX3 depletion of the Huh7.5.1 and HepG2/C3A cells, the recombinant retrovirus containing the shRNA against DDX3 was used. Briefly, GP2-293 cells were co-transfected with the plasmids pSIREN-RetroQ-ZsGreen-shDDX3 and pVSV-G at a ratio of 1:1. At 48 h later, the culture supernatant was harvested and subjected to centrifugation at 10,000× *g* for 2 min to remove the cell debris. The cleared supernatant was added to the Huh7.5.1 or HepG2/C3A cells, followed by incubation for 3 days.

### 2.6. BioID Assay

HEK293T cells in a 10-cm dish were transfected with Myc-BioID2-pORF2 or the empty vector Myc-BioID2-MCS. At 24 h later, the cells were treated with biotin at a final concentration of 50 µM. The cells were harvested with a lysis buffer (50 mM Tris HCl, pH 7.4, 500 mM NaCl, 0.2% SDS (*w*/*v*)), with the addition of the following before use: 1× protease inhibitor cocktail (Sigma-Aldrich, St. Louis, MO, USA) and 1 mM DTT at 24 h after the biotin treatment [32]. The cell lysate was subjected to dialysis against PBS pH 7.2 to eliminate the free biotin overnight. Then, the cell lysate was incubated with streptavidin-conjugated magnetic beads at 4 °C overnight. The beads were rinsed four times with the washing buffer, resuspended with 200 μL of PBS, incubated with biotin at a final concentration of 5 mM for 5 min before the addition of 2× Laemmli sample buffer, and heated at 100 °C for 10 min to elute the biotinylated proteins [33]. The elute was subjected to sodium dodecyl sulfate-polyacrylamide gel electrophoresis (SDS-PAGE) in-house and mass spectrometry analysis at the Taplin Mass Spectrometry Facility, Harvard Medical School (Boston, MA, USA). Briefly, the gel pieces were subjected to an in-gel trypsin digestion procedure [34]. Peptides were extracted, and high-performance liquid chromatography was conducted to elute the sample through a C18 column. The samples were run in an LTQ Orbitrap Velos Pro ion-trap mass spectrometer (Thermo Fisher Scientific, Waltham, MA, USA). The peptide sequences (and hence protein identity) were determined using a Sequest software (version 28) analysis (Thermo Fisher Scientific, Waltham, MA, USA) [35]. All the databases used included a reverse complement of all the sequences, and the data were filtered to between a one and two percent peptide false discovery rate. The total peptide counts were compared against 12 BioID control runs (HEK293) extracted from the CrapOME depository (http://www.crapome.org/ [36]). Proximal interactor hits were defined based on the following thresholds: detected with at least three unique peptides and a fold change greater than five times the average control runs (12 CrapOME + a study control) for each given ID (Appendix A).

### 2.7. Electroporation

The Huh7.5.1 cells in a 6-well plate were detached using a trypsin treatment, rinsed with Opti-MEM twice, and resuspended in 200 μL Opti-MEM. Full-length HEV RNA was added to the tubes with the cells at the amount of 4 μg RNA in each tube and mixed gently. The cell and RNA mixture was then added to a 0.4-cm cuvette, which was subjected to pulsation using Gene Pulser Xcell electroporation (Bio-Rad Laboratories, Hercules, CA, USA) at the conditions of 600 V, 1 pulse, and 0 intervals. After the electroporation, the cells were plated for culture.

### 2.8. Luciferase Reporter Assay and Cell Viability Assay

The samples of the cell culture supernatant were taken at 100 μL per well once every 24 h and frozen at −80 °C for further analysis. The culture wells were added with the same amount of fresh medium for further incubation. For the luciferase activity assay, the supernatant samples from the entire experiment were thawed together and tested using the Renilla luciferase assay system (Promega, Madison, WI, USA) according to the manufacturer’s protocol. The supernatant samples were added to a 96-well black flat-bottom microplate (Corning, Corning, NY, USA), followed by the addition of the same amount of substrate, and the reading of the luminescence signal was performed using a PerkinElmer 1420 Multilabel Counter (PerkinElmer Inc., Shelton, CT, USA). Three replicates were tested for each group. The CellTiter-Glo Luminescent Cell Viability Assay kit (Promega) was used to determine the cell viability following the manufacturer’s instructions.

### 2.9. Co-Immunoprecipitation (Co-IP) and Western Blotting

Co-IP was conducted with the antibodies against the FLAG-tag (Sigma-Aldrich, St. Louis, MO, USA) and the HEV capsid protein (homemade) as described [37]. Protein A/G Magnetic Beads for IP (Bimake, Houston, TX, USA) were used following the manufacturer’s instructions. The IP samples were subjected to a Western blot analysis with the antibodies against the Myc-tag, the HA-tag, and the capsid protein. SDS-PAGE and Western blotting were performed as described previously [27]. Antibodies against DDX3 (Proteintech), the HA-tag (Thermo Fisher, Waltham, MA, USA), the Myc-tag (Thermo Fisher), the FLAG-tag (Sigma-Aldrich), GAPDH (Santa Cruz Biotechnology, Inc., Dallas, TX, USA), and β-tubulin (Sigma-Aldrich) were used in the immunoblotting. The collection of the chemiluminescence signal was performed using the Quantity One Program, version 4.6 (Bio-Rad, Hercules, CA, USA), and a ChemiDoc XRS imaging system (Bio-Rad).

### 2.10. Reverse Transcription and Real-Time Quantitative PCR (RT-qPCR)

Total RNA was isolated from the GP2-293 cells with the TRIzol reagent (Thermo Fisher). Reverse transcription was performed with the Moloney murine leukemia virus reverse transcriptase (Thermo Fisher), oligo (dT), and a random 15-mer oligo. Quantitative PCR of the IFN-β and RPL32 (ribosomal protein L32, as an internal control of housekeeping gene) transcripts was performed using SYBR Green Supermix (Thermo Fisher) as described [27,37,38,39].

### 2.11. Computational Analysis of Protein–Protein Interaction

The structure of the HEV Kernow-C1 ORF2 (capsid) protein in dimeric form was generated using the homology modeling program Modeller [40], with a previously determined structure of the HEV capsid [41] (Protein Data Bank [42] code 2ZTN) used as a template. This structure and the previously determined structure of human DDX3 [43] (Protein Data Bank code 5E7I) were input into the ZDOCK server (http://zdock.umassmed.edu/, accessed on 8 October 2020) [44] to generate a model of the DDX3–capsid complex, with the default docking parameters used as the server input. The top-ranked ZDOCK server model (complex 1) is shown in Section 3.

### 2.12. Protein Network Analysis

The subcellular distribution of the biotinylated proteins from the mass spectrometry analysis was assessed using the ToppFun server (https://toppgene.cchmc.org/enrichment.jsp, accessed on 15 December 2020; Appendix A) and Cytoscape version 3.6.1 (https://cytoscape.org/index.html, accessed on 15 December 2020).

### 2.13. Statistical Analysis

Differences in the test indicators between the treated samples, such as the Gaussia luciferase level between DDX3-depleted cells and control cells, were assessed using the Student *t*-test. If a two-tailed *p* value is less than 0.05, it is considered significant.

## 3. Results

### 3.1. BioID and Mass Spectrometry Analysis

Our earlier study showed that the capsid protein inhibits type I interferon production via its arginine-rich motif (ARM) [27]. We hypothesized that other cellular factors might be involved in the inhibition process of IFN by the capsid protein. To assess this hypothesis, we performed a pilot BioID assay to gain insight on which host proteins are present in close proximity to the capsid protein. HEK293T cells were transfected with the Myc-BioID2-pORF2 or Myc-BioID2-MCS empty vector, and biotin was added for the biotinylation of the adjacent proteins. The biotinylation of the cellular proteins was verified using Western blotting. Comparing the pORF2-BioID data against our control and a set of CrapOME controls, we identified 145 potential proximal interactors (Appendix A; see Section 2). These hits were analyzed in ToppFun to identify the enriched GO Cellular Component categories (details in Appendix A). This analysis showed that the capsid protein was associated with the mitochondrion proteins (GO:0005739: 26/145; Figure 1A), suggesting that the capsid protein may interfere with some of the mitochondrial processes. This is consistent with our earlier finding that the capsid protein may be proximate to the mitochondrion as it interacts with MAVS and TBK1 to inhibit interferon induction [27]. Additionally, a great portion of the identified proteins were predicted to localize to ribonucleoprotein granules (GO:0035770: 24/145), supporting a role of the capsid protein in viral genome binding or the post-transcriptional regulation of cellular RNA. A GO biological process analysis revealed that ORF2 proximal interactors were highly enriched in proteins assigned to, e.g., RNA metabolism (GO:0016071: 28/145), the regulation of translation (GO:0006417: 20/145), and proteolysis (GO:0006508: 29/145) (Figure 1B).

### 3.2. The Capsid Protein Interacts with DDX3 Independent of RNA Association

The BioID/MS data analysis shows the top candidate with the highest sum intensity and number of peptides matched to DDX3 (Appendix A), suggesting a strong interaction with the capsid protein. To test the hypothesis, we further tested the interaction between DDX3 and the capsid protein using Co-IP. HEK293T cells were co-transfected with FLAG-DDX3 and Myc-BioID2-pORF2. Co-IP with an antibody against the FLAG-tag was conducted to precipitate DDX3, followed by Western blotting with Myc-tag antibody for the detection of the Myc-tagged capsid protein. The result showed that the capsid protein was present in the precipitates of DDX3, thus indicating a physical interaction (Figure 2A). Since both DDX3 and the capsid protein are RNA-associated proteins, we wondered whether the interaction between the two molecules was mediated by RNA. Co-IP was conducted in the presence or absence of RNase A. The result showed that the RNase treatment did not affect the co-precipitation of DDX3 and the capsid protein (Figure 2B), indicating that the interaction is independent of RNA association.

To further confirm the interaction between pORF2 and DDX3, we co-transfected HepG2/C3A, a liver-derived cell line, with plasmids encoding pORF2 and DDX3 and conducted an IFA to determine their co-localization. The IFA and confocal microscopy imaging results show that pORF2 and DDX3 have partial co-localization with a Pearson’s correlation coefficient (PCC) of 0.82 (Figure 2C). The result from the liver cells is consistent with the Co-IP finding, suggesting an interaction between these two proteins.

### 3.3. HEV Capsid Protein Interacts with the C-Terminal Region of DDX3

In order to map the domains that are responsible for the capsid protein and DDX3 interaction, the modeling of the complex structure was performed using the ZDOCK server [44]. The top-ranked model shows a possible interaction between the C-terminal domain of DDX3 and the C-terminal domain of the capsid protein dimer (Figure 3A), which corresponds to the “P” domain that protrudes from the surface of the full capsid assembly [41]. Two DDX3 truncations were constructed to test the prediction: N-terminal D1 and C-terminal D2 (Figure 3B). The DDX3 truncations and ORF2 plasmids were co-transfected into the HEK293T cells. Co-IP with the antibody against the capsid protein showed that DDX3-D2 but not DDX3-D1 was present in the precipitates (Figure 3C), which indicates that the C-terminal domain of DDX3 interacts with the capsid protein. The expression of DDX3-D1 was detected as doublet bands, although the lower band is at the expected size.

### 3.4. Depletion of DDX3 Impedes the IFN-β Induction

Since DDX3 enhances MAVS-mediated IFN signaling [45], we wondered whether DDX3 is involved in the capsid protein-mediated inhibition of IFN induction. In order to test this hypothesis, RNAi-mediated silencing of DDX3 in GP2-293 cells was conducted. Compared with the control shRNA, the shRNA against DDX3 (shDDX3) significantly reduced the DDX3 level (Figure 4A). The shRNA treatment had a minimal effect on the cell viability (Figure 4B). The cells were then transfected with the ORF2 plasmid and treated with poly(I:C). The expression of the IFN-β was then determined using RT-qPCR. The result showed that compared with the control cells, the depletion of DDX3 attenuated the IFN-β induction by over 50% (Figure 4C). The capsid protein inhibited IFN-β expression by 80% in control cells but 30% in DDX3-depleted cells. These results suggest that DDX3 may be involved in the capsid protein-mediated inhibition of IFN signaling.

### 3.5. DDX3 Depletion Reduces the Replication of HEV

In order to verify the role of DDX3 during HEV replication, Huh7.5.1 cells were transduced with a retrovirus containing shDDX3 or a control shRNA. The shDDX3-based gene silencing led to the depletion of DDX3 compared with the control shRNA (Figure 5A). A cell viability assay was performed when the cells were treated with shDDX3 for 3 days. The result showed that the DDX3 depletion had a minimal effect on the cell viability in comparison with the control shRNA (Figure 5B). The DDX3-depleted Huh7.5.1 cells were electroporated with the RNA of the HEV replicon p6/luc that encodes a secretory Guassia luciferase, an indicator of viral replication. The cell culture supernatant samples were collected every day for 8 consecutive days after the transfection. The luciferase assay result showed that HEV replication was significantly impaired by the depletion of DDX3, compared with the cells of the control shRNA (Figure 5C). The luciferase activity increased from day 3 and peaked at day 5 in the control cells, whereas the increase in the DDX3-silenced cells was quite limited.

In order to exclude the possibility of an off-target effect of the shRNA, the DDX3-silenced Huh7.5.1 cells were transfected with DDX3 plasmid to trans-compensate the expression. The WB result demonstrated the ectopic expression of DDX3 (Figure 5D). The DDX3-silenced Huh7.5.1 cells with DDX3 trans-compensation were transfected with the RNA of the HEV replicon p6/luc. The luciferase activity assay of the culture supernatant sample of day 5 after electroporation showed that the DDX3 reconstitution led to the restoration of most of the HEV replication, whereas the empty vector control failed to do so (Figure 5E).

The results above demonstrate the essential role of DDX3 in HEV replication, shown by an HEV replicon with luciferase. To further verify the role of DDX3 in the replication of wild-type HEV, we conducted DDX3 depletion in HepG2/C3A cells stably infected with the HEV Kernow-C1 strain. The WB result showed the DDX3 deficiency in the cells 3 days post-transduction with a retrovirus encoding shDDX3 (Figure 5F). The RT-qPCR result showed that the HEV RNA level in the DDX3-silenced cells was only 37% of that in the control cells (Figure 5G). These results demonstrated that DDX3 is extremely important in HEV replication.

### 3.6. The ATPase Motif of DDX3 Is Required for HEV Replication

Since both DDX3 ATPase and helicase activities are frequently involved in the protein functions, we wondered whether both the ATPase and helicase motifs are required for HEV replication. We constructed the ATPase mutant (K230E) and the helicase mutant (S382L) (Figure 6A) and transfected them into the DDX3-silenced Huh7.5.1 cells. Wild-type DDX3 plasmid and a YFP vector were included as controls. The WB result showed that the expression of the wild-type and the mutant DDX3 in the cells was at a similar level (Figure 6B). The DDX3-silenced Huh7.5.1 cells with the ectopic expression of wild-type or mutant DDX3 were electroporated with the RNA of the HEV replicon p6/luc. After the electroporation, culture supernatant samples were collected on day 5 for the luciferase assay. The result showed that, compared with the wild-type DDX3, the ATPase (K230E) mutant failed to restore the luciferase yield from the HEV replicon, whereas the helicase mutant (S382L) restored the yield (Figure 6C). The results indicate that the ATPase motif of DDX3 is extremely important in HEV replication.

## 4. Discussion

The HEV capsid protein is the major component of HEV virions, but its interaction with host proteins is less understood. The BioID/MS pilot analysis in this study identifies many putative interacting cellular partners, which link the viral protein to different subcellular compartments and a broad array of putative accessory functions, including RNA metabolism and protease activity. Consistent with our earlier findings, that the capsid protein inhibits MAVS-activated IFN induction, many putative interacting proteins localize to the mitochondrion, suggesting that the capsid protein may also affect the other mitochondrion-associated cellular activities. Many other interacting candidates are localized to the ribonucleoprotein granules with RNA-processing activities. Among the top candidates, DDX3 was validated to interact with the capsid protein using Co-IP. Notably, DDX3 was shown to be extremely important in HEV replication.

DDX3 is a member of the helicase family involved in cellular RNA regulation and is often involved in viral replication through either viral RNA transportation or translation [10]. In this study, the depletion of DDX3 was demonstrated to attenuate HEV replication. Given the putative functions of DDX3 in RNA metabolism, we postulated that its ATPase or helicase function might be needed for viral replication. Through the mutagenesis study, we showed that the mutant DDX3 with a K230E mutation failed to rescue the HEV replication in DDX3-depleted Huh7.5.1 cells, suggesting the involvement of the ATPase activity of DDX3 in the viral replication. The involvement of DDX3 in the viral replication may be mediated by a direct association of DDX3 and viral genome RNA, independent of its interaction with the capsid protein since the ORF2-null HEV replicon p6/luc was used. The expression of the luciferase in the replicon is from the subgenomic RNA, which is synthesized by HEV RdRp [29,46]. DDX3 is known to be involved in translational initiation and facilitate the newly assembled 80S ribosome [47,48]. DDX3 depletion possibly affects the translation of HEV RNA, leading to a low yield of RdRp and, consequently, little subgenomic RNA and luciferase production.

DDX3 synergizes with MAVS and TBK1 to enhance the production of IFNs [4,11]. Many viruses counteract this function. For instance, HCV’s core protein abolishes the IFN enhancement through its interaction with MAVS [45]. In this study, the depletion of DDX3 attenuated the production of IFN, indicating that the capsid protein’s negative regulation of IFN might also be via the binding of DDX3. The interaction between these two proteins is independent of RNA, although both are known to bind RNA. The HEV capsid protein inhibits the production of IFNs induced by poly (I:C), MAVS, and TBK1 through interacting with the MAVS-TBK1-IRF3 complex, and the presence of the capsid protein in the complex blocks the phosphorylation of IRF3 and the subsequent dissociation of IRF3 from the complex IRF3 [27]. The arginine-rich motif in the N-terminal domain of the capsid protein is needed to inhibit IRF3 phosphorylation. The interaction of DDX3 and the capsid protein potentially affects the function of the MAVS-TBK1-IRF3 complex, leading to a further attenuation of IFN induction. The modeling of the interaction of DDX3 with the capsid protein using the ZDOCK server suggests that the C-terminal domains of both proteins are involved. We verified the C-terminal domain of DDX3 in the interaction using IP, but we did not confirm the C-terminal domain of the capsid protein. This study reveals further mechanisms that might contribute to the capsid protein inhibition of interferon production.

DDX3 consists of two RecA-like domains. The C-terminal domain of DDX3 was found to be responsible for the interaction with the capsid protein. DDX3 domain 2 needs to rotate approximately 180° compared with domain 1 to obtain the closed conformation that is essential for RNA binding. The binding of the mononucleotide AMP helps to stabilize the DDX3 structure [49]. DDX3 contains nine highly conserved motifs of a DEAD-box family RNA helicase: motifs Q, I, II, Ia, Ib, and III are located in domain 1, and motifs IV, V, and VI are found in domain 2. Structural, biochemical, and mutational analyses demonstrate that the different motifs are critical in various processes, such as nucleotide binding (Q, I, and II), RNA binding (Ia, Ib, IV, and V), and ATP hydrolysis (III and possibly VI) [2]. DDX3 interacts directly with dsRNA through the N-terminal D1 domain. The contact residues are G302 and G325, and the engagement of RNA to DDX3 helps to trigger the ATPase activity of DDX3 [50]. After the ATP binding of the DDX3:dsRNA complex, DDX3 unwinds the RNA duplex, resulting in a complex composed of DDX3, single-stranded RNA (ssRNA), and ATP. ATP hydrolysis then facilitates the release of ssRNA [51,52]. The capsid protein interaction of domain 2 of DDX3 might affect its RNA binding.

The HEV capsid protein contains three major domains: S, M, and P [53]. After synthesis, the capsid protein forms dimers and further assembles into icosahedral virus-like particles (VLPs). The crystal structure of the VLP shows that all of the three domains have some area exposed to the surface of the VLP, with the P domain protruding outside [53]. A total of 30 dimeric spikes are situated on the icosahedral 2-fold symmetry axes. Our computational model predicted that DDX3 and the capsid protein interact via their C-terminal domains. The construct of DDX3-D1 was expressed as doublet bands shown in Figure 3C, although the lower band is at the expected size. The reason for the presence of the top band is unknown. This was not further pursued since DDX3-D1 did not co-precipitate with the capsid protein. The co-IP result showed that the C-terminal domain of DDX3 is the predominant interaction site. This interaction is not required for HEV replication as the K230 ATPase motif at the N-terminal domain of DDX3 is extremely important for viral proliferation. These results indicate that DDX3 plays a pro-viral role in HEV replication, while its assistance of IFN induction is blocked by the capsid protein. Further research is warranted to elucidate the mechanism of DDX3 involvement in HEV infection.

DDX3 is known to be required for some viral replication [12]. During HCV replication, the DDX3 C-terminal domain interacts with the N-terminal 59 residues of the HCV core protein independent of RNA [54]. Moreover, DDX3 is recruited to the lipid droplets, where HCV virions assemble, and the depletion of DDX3 in hepatoma cells significantly inhibits HCV replication [13]. A later study shows that a single amino acid mutation within the N-terminal domain of the HCV core protein abrogates its interaction with DDX3 without affecting the virus propagation [55]. After the mutation of the core protein gene in the virus genome, DDX3 no longer co-localizes with the core protein in HCV-replicating hepatoma cells. However, the silencing of DDX3 still results in the inhibition of HCV replication, indicating that the interaction between DDX3 and the core protein is not required for viral replication, and the pro-viral role of DDX3 is independent of its subcellular translocation [55]. Another report demonstrated that DDX3 augments the MAVS-mediated increase in IFN production through direct interaction with MAVS [11]; while this promotion can be counteracted by the HCV core protein, possibly due to the hijack of DDX3 by the core protein from the RIG-I/MAVS/DDX3 complex to lipid droplets [11], DDX3 also contributes to the viral replication of the Japanese encephalitis virus (JEV). The absence of DDX3 leads to defective viral replication without affecting viral assembly and release. The overexpression of the DDX3 with an ATPase motif mutation, K230E, does not rescue viral replication, indicating a critical role of the ATPase motif of DDX3. Additionally, DDX3 co-localizes with viral NS3, NS5, and dsRNA, and interacts with 5′UTR and 3′UTR to facilitate viral replication [56].

Limitations of the study: Co-IP was performed for the lysate of HEK293 cells instead of HEV-infected liver cells. To address this, we conducted an IFA of liver-derived HepG2/C3A cells to show the co-localization of pORF2 and DDX3. Also, the mechanism of the contribution of DDX3 to HEV replication is not known, as the interaction between the capsid protein and DDX3 is dispensable. Future studies are needed to address which step of HEV replication involves DDX3.

## 5. Conclusions

In conclusion, the HEV capsid protein interacts with DDX3 to inhibit its role in IFN induction, and the virus takes advantage of DDX3 for efficient replication. The ATPase motif of DDX3 is involved in HEV replication. These results reveal that the HEV regulation of DDX3 contributes to a conducive environment of replication in different aspects, including viral genome replication and innate immunity inhibition. This study provides further insights on virus–cell interactions.

## Figures and Tables

**Figure 1 pathogens-14-00177-f001:**
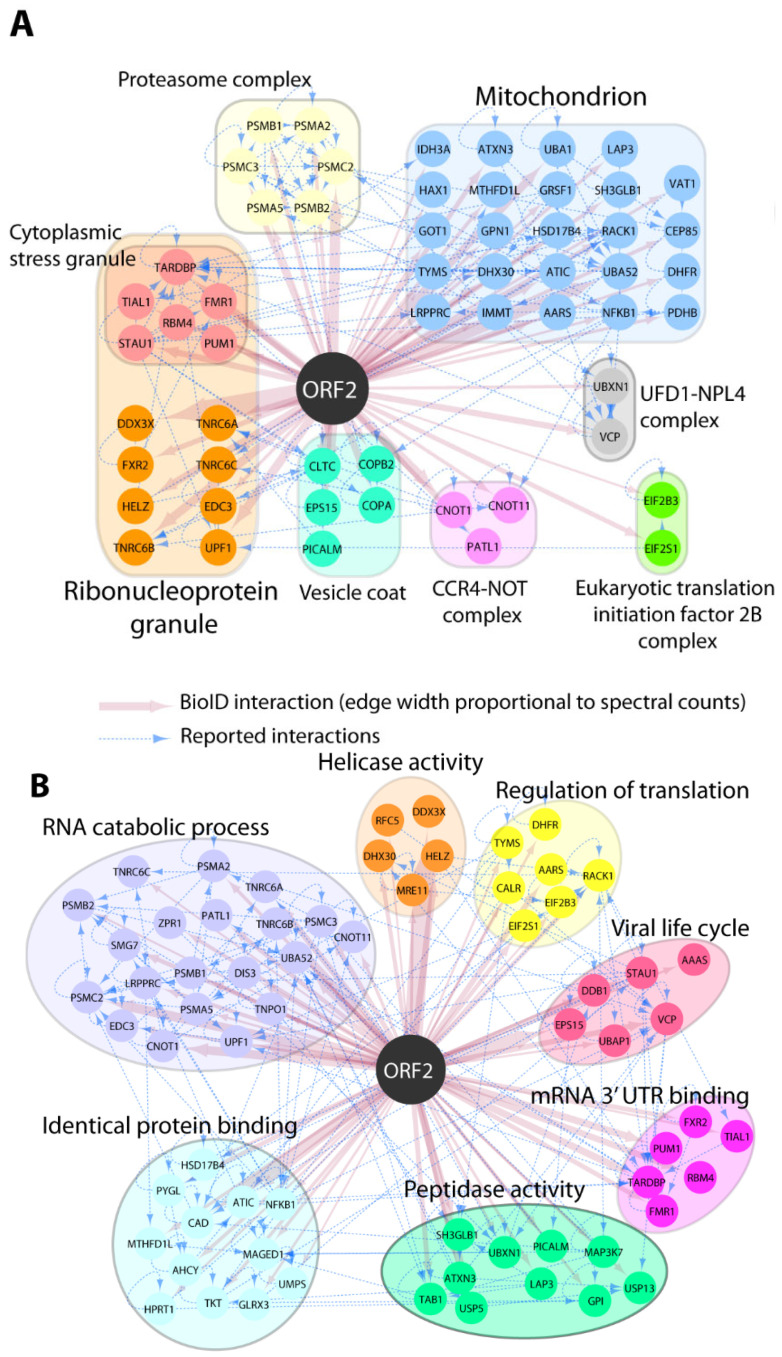
GO Enrichment of pORF2-proximal proteins. The selection of pORF2 interactors is indicated by color-coded nodes and grouped according to their intracellular localization (GO CC; (**A**)) and functions (GO BP; (**B**)). Selected hits and corresponding relevant categories are indicated. BioID interactions are depicted in red, and reported interactions between hits (collected from http://iid.ophid.utoronto.ca/ (accessed on 10 December 2020); using experimental evidence as a threshold; sources detailed in Appendix A) are shown by dashed blue edges.

**Figure 2 pathogens-14-00177-f002:**
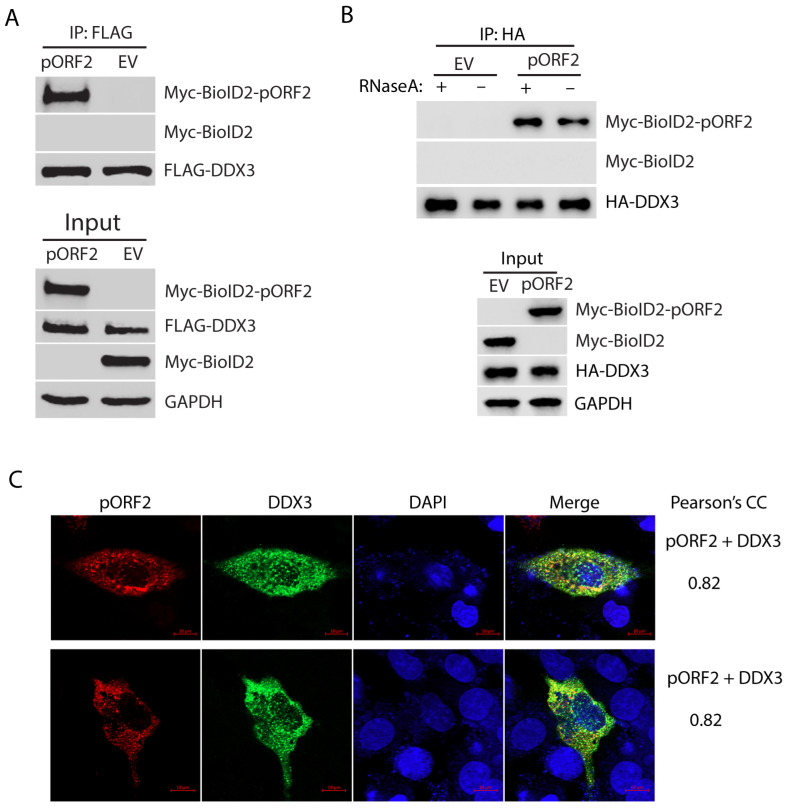
DDX3 co-precipitates the capsid protein, independent of RNA association. (**A**) Co-IP of DDX3 precipitates the capsid protein. HEK293T cells were co-transfected with FLAG-DDX3 and Myc-BioID2-pORF2 and harvested for Co-IP with the FLAG-tag antibody at 36 h post-transfection (hpt). The IP precipitate and input were subjected to Western blotting (WB). A Myc-BioID2 empty vector (EV) was used as a negative control. Original Western blot images can be found in Appendix A and S2. (**B**) The co-precipitation of DDX3 and the capsid protein is independent of RNA. HEK293T cells were co-transfected with HA-DDX3 and Myc-BioID2-pORF2. The cells were harvested 24 hpt. The cell lysate was treated with RNase A at 333 μg/mL for 20 min at room temperature, followed by Co-IP and WB. Original Western blot images can be found in Appendix A. (**C**) IFA and confocal microscopy show the co-localization of pORF2 (red) and DDX3 (green). HepG2/C3A cells were co-transfected with Myc-BioID2-pORF2 and FLAG-DDX3 for 42 h. The cells were fixed and probed with a rabbit anti-Myc-tag antibody and a mouse anti-FLAG antibody. The bars in the lower right corners of the images denote 10 μm. The Pearson’s CC is indicated on the right.

**Figure 3 pathogens-14-00177-f003:**
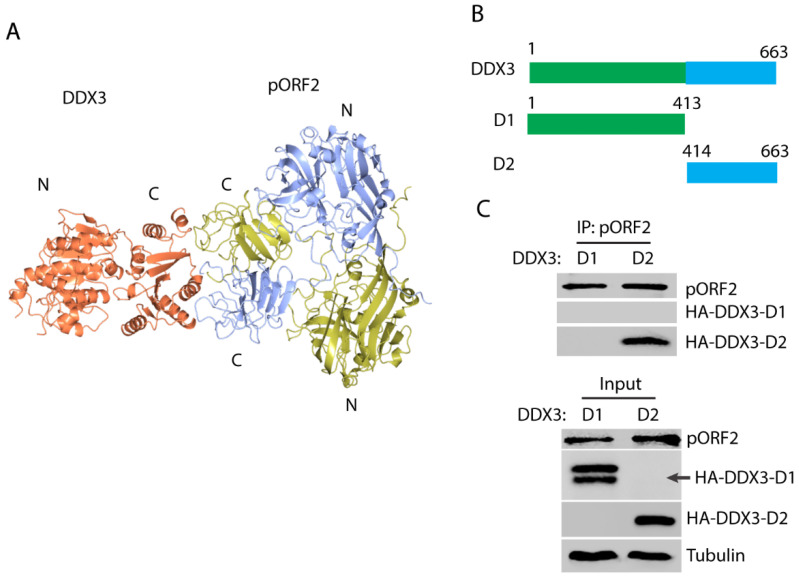
The C-terminal domain of DDX3 interacts with the capsid protein. (**A**) A model of the interaction between DDX3 and the capsid protein. The capsid protein is in a homodimeric form and the chains are colored yellow and blue, and DDX3 is colored orange. “N” stands for the N-terminus and “C” for the C-terminus; these are labeled for each protein. (**B**) A schematic illustration of the two DDX3 domains and the truncations. The numbers above the lines indicate amino acid residues of DDX3. (**C**) The capsid protein co-precipitates the C-terminal domain of DDX3. The HEK293T cells were transfected with plasmids of DDX3-D1, D2, and pORF2. Co-IP with the antibody against pORF2 was conducted, followed by Western blotting with an antibody against the HA-tag for DDX3. Cell lysate input was included in the Western blotting as a control. Doublet bands were detected for HA-DDX3-D1, and its expected size is indicated by an arrow. Original Western blot image can be found in Appendix A.

**Figure 4 pathogens-14-00177-f004:**
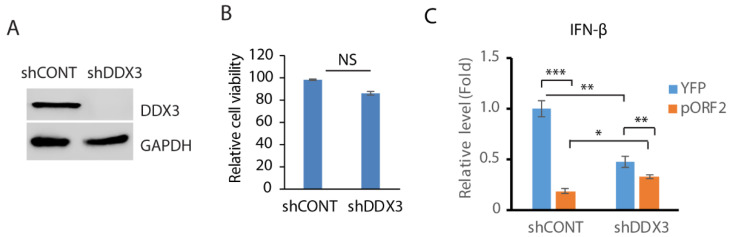
Depletion of DDX3 leads to the inhibition of IFN-β expression. (**A**) Depletion of DDX3 in GP2-293 cells. The cells were transfected with pSIREN-RetroQ-ZsGreen-shDDX3 and pVSV-G plasmids for 3 days and were then harvested for WB. (**B**) DDX3 silencing has a minimal effect on GP2-293 cell viability. The cells were co-transfected with pSIREN-RetroQ-ZsGreen-shDDX3 and pVSV-G plasmids for 3 days and were then harvested for the cell viability assay. Three replicates were used for each sample. NS: no significant difference. (**C**) DDX3 depletion leads to a significant reduction in poly(I:C)-induced interferon-β expression. DDX3-depleted GP2-293 cells were transfected with the capsid protein or the YFP plasmid for 24 h and then transfected with poly(I:C) overnight. The cells were harvested for RNA extraction and RT-qPCR for IFN-β expression. Three replicates were used for each sample. The transcript of the housekeeping gene RPL32 was also determined for normalization. Relative levels in comparison with the YFP and shRNA control are shown. *** denotes *p* < 0.001, ** for *p* < 0.01 and * for *p* < 0.05. Original Western blot image can be found in Appendix A.

**Figure 5 pathogens-14-00177-f005:**
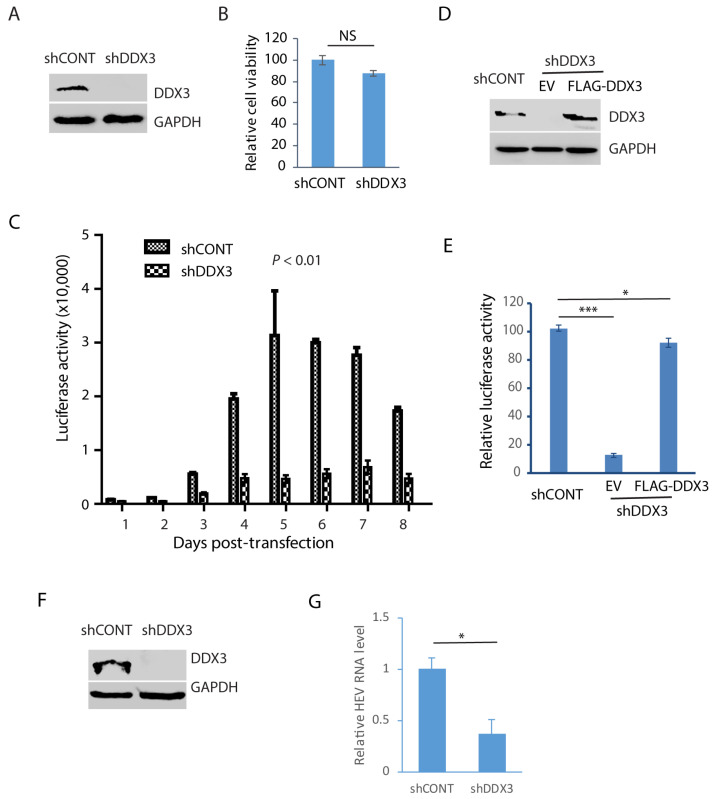
Depletion of DDX3 results in a significant reduction in HEV replication. (**A**) Western blot detection of DDX3 in Huh7.5.1 cells after the DDX3 depletion. The cells were transduced with a retrovirus containing DDX3-shRNA (shDDX3) or control shRNA (shCONT) for 3 days and were harvested for WB detection of DDX3. (**B**) Depletion of DDX3 has a minimal effect on the cell growth. The Huh7.5.1 cells were transduced with a retrovirus containing shDDX3 for 3 days before the cell viability was tested. Three replicates were used in each group. NS: no significant difference. (**C**) DDX3 depletion reduces the luciferase yield from the HEV replicon. Huh7.5.1 cells with or without DDX3 silencing were electroporated with the RNA of the p6/luc replicon. The daily culture supernatant samples were assayed for luciferase activity. (**D**) WB of DDX3 trans-compensation in DDX3-depleted Huh7.5.1 cells. The cells were transduced with a retrovirus encoding shDDX3 for 2 days and then transfected with FLAG-DDX3 or EV (empty vector) for another 2 days. (**E**) Ectopic expression of DDX3 restores most HEV replication. The Huh7.5.1 cells with DDX3 reconstitution were electroporated with the RNA of the HEV replicon p6/luc. The luciferase activity in the culture supernatant samples was tested 5 days later. * denotes *p* < 0.05, *** denotes *p* < 0.001. (**F**) Depletion of DDX3 in HepG2/C3A cells stably infected with the HEV Kernow-C1 strain. The cells were transduced with a retrovirus encoding shDDX3 or shCONT. (**G**) HEV RNA level in DDX3-depleted HepG2/C3A cells. The Kernow-infected cells were harvested for RNA isolation 3 days post-shDDX3 treatment. The transcript of RPL32 was also determined for normalization. The relative levels in comparison with shCONT are shown. * denotes *p* < 0.05. Original Western blot images can be found in Appendix A.

**Figure 6 pathogens-14-00177-f006:**
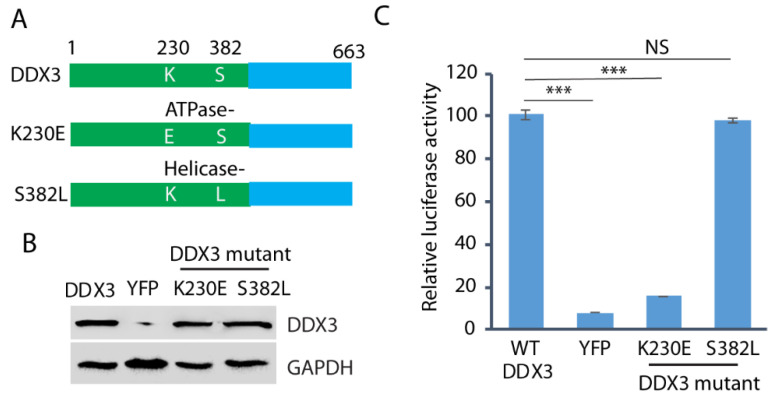
The ATPase motif of DDX3 is extremely important in HEV replication. (**A**) Schematic illustration of DDX3 and its mutants. The numbers above the lines indicate the amino acid residues of DDX3. (**B**) WB detection of the protein expression of the DDX3 and mutants in DDX3-silenced Huh7.5.1 cells. The Huh7.5.1 cells were transduced with a retrovirus encoding shDDX3 for 2 days and then transfected with DDX3, YFP vector, or mutant DDX3 (K230E and S382L) plasmids for another 2 days. (**C**) Restoration of HEV replication in cells trans-compensated with the DDX3 wild-type and the helicase mutant. Two days after the reconstitution with the wild-type or mutant DDX3, the Huh7.5.1 cells were electroporated with the RNA of the HEV replicon p6/luc. The luciferase activity in the samples of the culture supernatant was determined on day 5. Relative luciferase activity is shown. NS: no significant difference; ***: *p* < 0.001. Original Western blot image can be found in Appendix A.

**Table 1 pathogens-14-00177-t001:** Primers used in this study.

Primer ^a^	Sequence (5′ to 3′) ^b^	Target
KH2F4	G*CTCGAG*TGCCCTAGGGTTGTTCTGCTGCTGTTC	Kernow ORF2
KH2R4	ACGTCTC*GAATTC*TTAAGACTCCCGGGTTTTGCCTACCTCCG	Kernow ORF2
DDX3-F1	C*GAATTC*AGTCATGTGGCAGTGGAAAAT	DDX3
DDX3-R1	A*CTCGAG*TCAGTTACCCCACCAGTCAAC	DDX3
DDX3-D1-F1	T*GAATTC*GCCACCATGAGTCATGTGGCAGTGGA	DDX3 D1
DDX3-D1-R1	GCTCGAGTTCAGAGGTAGAGCCAACTCT	DDX3 D1
DDX3-D2-F2	TGAATTCGCCACCATGGCTGTAGGAAGAGTTGGCTC	DDX3 D2
DDX3-D2-R2	GCTCGAGGTTACCCCACCAGTCAACC	DDX3 D2
shDDX3-F1	GATCCGGAGTGATTACGATGGCATTGTTCAAGAGACAATGCCATCGTAATCACTCCTTTTTTG	DDX3
shDDX3-R1	AATTCAAAAAAGGAGTGATTACGATGGCATTGTCTCTTGAACAATGCCATCGTAATCACTCCG	DDX3
DDX3-K230E-F1	TGTGCCCAAACAGGGTCTGGAGAGACTGCAGCATTTCTGTTGCCC	DDX3 K230E
DDX3-K230E-R1	GGGCAACAGAAATGCTGCAGTCTCTCCAGACCCTGTTTGGGCACA	DDX3 K230E
DDX3-S382L-F1	TTCCTTAGGAAAAGTAGCCAAAAACATCATAGTGTGGCG	DDX3 S382L
DDX3-S382L-R1	CGCCACACTATGATGTTTAGTGCTACTTTTCCTAAGGAA	DDX3 S382L

^a^ F: forward primer, R: reverse primer. The ORF2 sequence is from HEV Kernow-C1 P6 (GenBank accession# JQ679013). ^b^ The italicized alphabets indicate the restriction enzyme cleavage sites for cloning.

## Data Availability

Data are presented in this manuscript.

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
