# Peer review of "RNA Helicase DDX3 Interacts with the Capsid Protein of Hepatitis E Virus and Plays a Vital Role in the Viral Replication"

_pathogens, 2025, doi:10.3390/pathogens14020177_

Round 1
Reviewer 1 Report
Comments and Suggestions for Authors
As the causative agent, Hepatitis E virus (HEV) poses a severe threat to the human and animal health. Lin et al in their manuscript titled “ RNA helicase DDX3 interacts with the capsid protein of 2 hepatitis E virus and plays a vital role in the viral replication” investigate on the role of DDX3, an ATP-dependent RNA helicase, in the replication of the hepatitis E virus (HEV). Through bioinformatic, Co-IP, protein truncation, gene knockdown and other approaches, they demonstrated that DDX3 interact with the HEV capsid protein and play an extremely important role in HEV 17 replication. They also defined the domain and motifs of DDX3 involved in interacting with the capsid protein by multiple approaches. The results enrich the current understanding on HEV pathogenesis and provide insights into the HEV interaction with host to beneficial to virus replication. Generally, the experiments are well-designed, and manuscript is well-written. The data is also informative to the readers and peers .
Several minor modifications suggested
1. In line 58-60, the authors should add the overall genotype information of HEV (HEV-1~8) and their host spectrum followed by emphasizing the Genotype 1,2,3, and 4.
2. Line 360-361, change the sentence to “The results indicate that the ATPase motif of DDX3 is extremely important for HEV replication”
3. In line 368-369, change the subtitle of the figure legend to “C. Restoration of HEV replication in cells trans-compensated with the DDX3 wild-type and the helicase-mutant”
4. Line 387: change sentence “In this study, the depletion of DDX3 attenuates HEV replication” to “In this study, the depletion of DDX3 was demonstrated to attenuate HEV replication”
Author Response
Reviewer 1
As the causative agent, Hepatitis E virus (HEV) poses a severe threat to the human and animal health. Lin et al in their manuscript titled “ RNA helicase DDX3 interacts with the capsid protein of 2 hepatitis E virus and plays a vital role in the viral replication” investigate on the role of DDX3, an ATP-dependent RNA helicase, in the replication of the hepatitis E virus (HEV). Through bioinformatic, Co-IP, protein truncation, gene knockdown and other approaches, they demonstrated that DDX3 interact with the HEV capsid protein and play an extremely important role in HEV 17 replication. They also defined the domain and motifs of DDX3 involved in interacting with the capsid protein by multiple approaches. The results enrich the current understanding on HEV pathogenesis and provide insights into the HEV interaction with host to beneficial to virus replication. Generally, the experiments are well-designed, and manuscript is well-written. The data is also informative to the readers and peers .
Thank you for the comments.
Several minor modifications suggested
- In line 58-60, the authors should add the overall genotype information of HEV (HEV-1~8) and their host spectrum followed by emphasizing the Genotype 1,2,3, and 4.
Thank you for the comments. The information is added: “HEV strains are heterogenic and there are four genera in the subfamily Orthohepevirinae, the Hepeviridae family [21]. There are eight genotypes in the species Paslahepevirus balayani, the genus Paslahepevirus. At least four of the eight genotypes can infect humans: genotypes 1 and 2 are restricted to humans, whereas genotypes 3 and 4 are zoonotic with an expanded host range, including monkey, pig, sheep, cow, wild boar, deer, rabbit, and mongoose [22,23]. Genotypes 5 and 6 are isolated from wild boars [24] and genotypes 7 and 8 are isolated from camels [25,26]. Strains of genotypes 5 and 7 are suspected of the potential for zoonotic infection [26,27].”
- 2.Line 360-361, change the sentence to “The results indicate that the ATPase motif of DDX3 is extremely important for HEV replication”
Thank you for the comments. Done as suggested.
- 3.In line 368-369, change the subtitle of the figure legend to “C. Restoration of HEV replication in cells trans-compensated with the DDX3 wild-type and the helicase-mutant”
Thank you for the comments. Done as suggested.
- Line 387: change sentence “In this study, the depletion of DDX3 attenuates HEV replication” to “In this study, the depletion of DDX3 was demonstrated to attenuate HEV replication”
Thank you for the comments. Done as suggested.
Reviewer 2 Report
Comments and Suggestions for Authors
This manuscript entitled “RNA helicase DDX3 interacts with the capsid protein of hepatitis E virus and plays a vital role in the viral replication” by Lin et al., reports a pro-viral role of an ATP-dependent RNA helicase DDX3 in HEV replication. The authors identified DDX3 as an interacting protein of HEV ORF2 by BioID-MS and Co-IP. Depletion of DDX3 leads to the inhibition of IFN-β expression and the reduction of HEV replication, while the ATPase motif of DDX3 is extremely important for HEV replication. I have several concerns which need to be addressed
Comments:
1. As the author revealed that the knockdown DDX3 inhibits HEV replication via an ATPase motif of DDX3 dependent manner. But what is the mode of mechanism of DDX3? Could DDX3 unwind HEV RNA to promote its replication? Or is it via the interaction between ORF2 and DDX3? Moreover, As the author described in the introduction, DDX3 couples with mitochondrial antiviral-signaling protein (MAVS) to promote the transcription of interferons (IFNs). Depletion of DDX3 reduces IFN production, which in turn will promote HEV replication. Then what is the role of DDX3 on IFN and HEV in your experimental system? The authors need to clarify all these remaining questions to draw a clear conclusion.
2. ORF2 binds to the C-terminal region of DDX3, but which region of ORF2 binds to DDX3 still needs to be investigated.
3. The authors observed a reduction of IFN-β mRNA after knockdown of DDX3 in fig4, whether there is a change in the protein level of IFN-β and downstream interferon-stimulated genes needs to be further demonstrated, and does overexpression of DDX3 affects IFN-β expression inhibited by ORF2?
4. In fig.6, The mutations in ATPase and helicase motif of DDX3 affect the replication of HEV replicon p6/luc. This observation should be further validated in the full-length HEV clones.
Author Response
Reviewer 2
This manuscript entitled “RNA helicase DDX3 interacts with the capsid protein of hepatitis E virus and plays a vital role in the viral replication” by Lin et al., reports a pro-viral role of an ATP-dependent RNA helicase DDX3 in HEV replication. The authors identified DDX3 as an interacting protein of HEV ORF2 by BioID-MS and Co-IP. Depletion of DDX3 leads to the inhibition of IFN-β expression and the reduction of HEV replication, while the ATPase motif of DDX3 is extremely important for HEV replication. I have several concerns which need to be addressed
Thank you for the comments.
Comments:
- As the author revealed that the knockdown DDX3 inhibits HEV replication via an ATPase motif of DDX3 dependent manner. But what is the mode of mechanism of DDX3? Could DDX3 unwind HEV RNA to promote its replication? Or is it via the interaction between ORF2 and DDX3? Moreover, As the author described in the introduction, DDX3 couples with mitochondrial antiviral-signaling protein (MAVS) to promote the transcription of interferons (IFNs). Depletion of DDX3 reduces IFN production, which in turn will promote HEV replication. Then what is the role of DDX3 on IFN and HEV in your experimental system? The authors need to clarify all these remaining questions to draw a clear conclusion.
Thank you for the comments. We discussed the possible DDX3 role in HEV replication in Discussion. HEV appears to use DDX3 ATPase activity for efficient replication while inhibiting the DDX3 role in interferon induction via the capsid protein. Since DDX3 is a multi-functional protein, the virus deals with it with multiple strategies. We updated the conclusion to reflect this point. This is like HCV needing DDX3 for RNA replication but HCV core protein interaction with DDX3 is dispensable. The involvement of DDX3 in HEV replication may be mediated by a direct association of DDX3 and viral genome RNA independent of its interaction with the capsid protein since the ORF2-null HEV replicon p6/luc was used. The expression of the luciferase in the replicon is from the subgenomic RNA, which is synthesized by HEV RdRp. DDX3 is known to be involved in translational initiation and facilitate the newly assembled 80S ribosome. DDX3 depletion possibly affects the translation of HEV RNA, leading to a low yield of RdRp and, consequently, little subgenomic RNA and luciferase production. On the other hand, DDX3 synergizes with MAVS and TBK1 to enhance the production of IFN. Many viruses counteract this function of DDX3. For instance, HCV's core protein abolishes the IFN enhancement through interaction with DDX3. Our data demonstrate that the depletion of DDX3 attenuated the production of IFN, indicating the capsid protein’s negative regulation of IFN might also be via binding DDX3.
- ORF2 binds to the C-terminal region of DDX3, but which region of ORF2 binds to DDX3 still needs to be investigated.
Thank you for the comments. The modeling of the interaction of DDX3 with the capsid protein with the ZDOCK server suggests that the C-terminal domains of both proteins are involved. We verified the C-terminal domain of DDX3 in the interaction by IP, but we did not confirm the C-terminal domain of the capsid protein. Future study is needed to investigate this interaction on the capsid protein side.
- The authors observed a reduction of IFN-β mRNA after knockdown of DDX3 in fig4, whether there is a change in the protein level of IFN-β and downstream interferon-stimulated genes needs to be further demonstrated, and does overexpression of DDX3 affects IFN-β expression inhibited by ORF2?
Thank you for the comments. The Fig 4 shows the reduction of IFN-β mRNA after knockdown of DDX3. It is expected the protein level of IFN-β and downstream interferon-stimulated genes will be reduced. We did not detect these proteins, since it was verified in previous publications. Overexpression of DDX3 will not affect much of pORF2-mediated inhibition of IFN because we demonstrated previously that the capsid protein blocks the IRF3 phosphorylation via interaction with the complex of MAVS, TBK1 and IRF3 and that the N-terminal motif is essential for the inhibition. The pORF2 interaction with DDX3 may enhance the inhibition but may be dispensable.
- In fig.6, The mutations in ATPase and helicase motif of DDX3 affect the replication of HEV replicon p6/luc. This observation should be further validated in the full-length HEV clones.
Thank you for the comments. It is expected DDX3 ATPase mutant will affect the replication of full-length HEV because the HEV replicon p6/luc was established to study HEV replication and the luciferase expression is from the subgenomic RNA that is produced only from HEV RNA replication. The results from full-length HEV clones will validate the finding but won’t change the conclusion. Future study is needed to investigate further mechanisms.
Reviewer 3 Report
Comments and Suggestions for Authors
In the present study, the authors investigate the role of DDX3 in the replication of HEV. Using a BioID assay, DDX3 was identified as a molecule interacting with ORF2. Immunoprecipitation demonstrated that ORF2 interacts with the C-terminal region of DDX3, and this interaction is RNA-independent. Knockdown of DDX3 attenuated the suppression of IFN-beta production by ORF2. Additionally, experiments using a replicon expressing luciferase instead of ORF2 showed that DDX3 is involved in HEV replication. With DDX3 mutants, they revealed that the ATPase activity of DDX3 affects HEV replication.
The authors demonstrate that ORF2 influences interferon production through DDX3 and that the ATPase activity of DDX3 is essential for HEV replication. While these two significant findings are reasonably well supported, the authors need to select appropriate cell lines for their experiments and conduct additional experiments to support their results.
Comments:
1. HEK293T-derived cells are unsuitable for use in experiments as they cannot support HEV replication. Specifically, for experiments evaluating the impact on IFN-beta production, the authors should be used liver-derived cell lines.
2. In addition to immunoprecipitation, in situ data such as proximity ligation assays are necessary to confirm the interaction between ORF2 and DDX3. At a minimum, co-localization should be demonstrated under a microscope. These experiments need to be conducted in liver-derived cell lines. These studies should be performed in HEV-infected cells to observe the interaction between HEV-derived ORF2 and endogenous DDX3.
3. The expression of interferon-β should be determined at the protein level using ELISA, rather than at the mRNA level. This experiment needs to be conducted using liver-derived cell lines.
4. In figure 2A, the Myc-BioID2-pORF2 band appears to be undetected in the original figure.
5. In figure 3A, it appears that this figure has been mixed up with another one.
6. In figure 3C, it appears that HA-tagged ORF2 is being used, but it is not mentioned in the Materials and Methods section.
Author Response
Reviewer 3
In the present study, the authors investigate the role of DDX3 in the replication of HEV. Using a BioID assay, DDX3 was identified as a molecule interacting with ORF2. Immunoprecipitation demonstrated that ORF2 interacts with the C-terminal region of DDX3, and this interaction is RNA-independent. Knockdown of DDX3 attenuated the suppression of IFN-beta production by ORF2. Additionally, experiments using a replicon expressing luciferase instead of ORF2 showed that DDX3 is involved in HEV replication. With DDX3 mutants, they revealed that the ATPase activity of DDX3 affects HEV replication.
The authors demonstrate that ORF2 influences interferon production through DDX3 and that the ATPase activity of DDX3 is essential for HEV replication. While these two significant findings are reasonably well supported, the authors need to select appropriate cell lines for their experiments and conduct additional experiments to support their results.
Thank the reviewer for the comments.
Comments:
- HEK293T-derived cells are unsuitable for use in experiments as they cannot support HEV replication. Specifically, for experiments evaluating the impact on IFN-beta production, the authors should be used liver-derived cell lines.
Thank the reviewer for the comments. We agree with the reviewer that it is best to use liver-derived cell lines for HEV study, but not ideal for IFN study because HEV293T cells are frequently used for easy transfection. Also, it is fine to study the effect of individual proteins on IFN production in HEK293T cells, as shown in studies in the IFN field.
- 2.In addition to immunoprecipitation, in situ data such as proximity ligation assays are necessary to confirm the interaction between ORF2 and DDX3. At a minimum, co-localization should be demonstrated under a microscope. These experiments need to be conducted in liver-derived cell lines. These studies should be performed in HEV-infected cells to observe the interaction between HEV-derived ORF2 and endogenous DDX3.
Thank the reviewer for the comments. We agree that proximity ligation assays can confirm the interaction between ORF2 and DDX3. Observing colocalization with the capsid protein is supplemental to the Co-IP result but will not change the conclusion. Co-IP results show the capsid protein co-precipitates DDX3-D2 but not DDX3-D1 in Figure 3. The results in both Figures 2 and 3 indicate the interaction between DDX3 and the capsid protein. Further experiments will confirm the findings. It is expected that the two proteins will interact in liver-derived cell lines although we did not conduct the test because transfection of such cell lines is much more difficult than HEK293T cells. Future study will be conducted to observe the interaction in HEV-infected cells when examining the mechanism of the interaction in HEV infection.
- 3.The expression of interferon-β should be determined at the protein level using ELISA, rather than at the mRNA level. This experiment needs to be conducted using liver-derived cell lines.
Thank the reviewer for the comments. We agree that protein level detection of interferon is a stringent proof, and it is expected DDX3 depletion will reduce IFN production. Using liver-derived cell lines will verify the finding but won’t change the conclusion. We believe that the results are sufficient to show the effect of DDX3 depletion on IFN-β expression.
- 4.In figure 2A, the Myc-BioID2-pORF2 band appears to be undetected in the original figure.
We double-checked that the Myc-BioID2-pORF2 band is clearly shown in the original image of Figure 2. Please double check.
- In figure 3A, it appears that this figure has been mixed up with another one.
Thanks. We have updated the figure with the correct model for figure 3A.
- In figure 3C, it appears that HA-tagged ORF2 is being used, but it is not mentioned in the Materials and Methods section.
Thanks. We have updated the Materials and Methods section to include the HA-tagged ORF2.
Round 2
Reviewer 3 Report
Comments and Suggestions for Authors
In this study, the authors demonstrated that DDX3, which interacts with ORF2, is involved in the mechanism by which ORF2 suppresses interferon-β production in HEK293-derived cells. Independently, they also showed that the ATPase activity of DDX3 is involved in HEV replication.
Thank you for making the revisions according to the comments. However, their response is still insufficient.
The interaction between ORF2 and DDX3 has only been examined under artificial conditions using co-IP in HEK293-derived cells. Therefore, validation using liver-derived cells under physiological conditions is essential. The authors should at least confirm co-localization under a microscope. If the interaction cannot be confirmed under physiological conditions, the results obtained from co-IP may represent non-specific binding.
Author Response
In this study, the authors demonstrated that DDX3, which interacts with ORF2, is involved in the mechanism by which ORF2 suppresses interferon-β production in HEK293-derived cells. Independently, they also showed that the ATPase activity of DDX3 is involved in HEV replication.
Thank you for making the revisions according to the comments. However, their response is still insufficient.
The interaction between ORF2 and DDX3 has only been examined under artificial conditions using co-IP in HEK293-derived cells. Therefore, validation using liver-derived cells under physiological conditions is essential. The authors should at least confirm co-localization under a microscope. If the interaction cannot be confirmed under physiological conditions, the results obtained from co-IP may represent non-specific binding.
Thank the reviewer for the comment. We have conducted immunofluorescence assay of liver-derived HepG2/C3A cells and added the images showing the co-localization of ORF2 and DDX3 (Figure 2C). IFA and confocal microscopy imaging results show that pORF2 and DDX3 have partial co-localization with Pearson’s correlation coefficient (PCC) of 0.82 (Figure 2C). The result from liver cells is consistent with the Co-IP finding in HEK-293 cells, suggesting the interaction between these two proteins.
Round 3
Reviewer 3 Report
Comments and Suggestions for Authors
Thank you for the additional experiments and manuscript revisions. I believe it is now suitable for publication in its current form.